# Size control and oscillations of active droplets in synthetic cells

Judit Sastre[1], Advait Thatte[2], Alexander M. Bergmann[1], Michele Stasi[1], Marta Tena-Solsona[1], Christoph A. Weber [2] ✉ & Job Boekhoven [1] ✉

Oscillations in the formation and dissolution of molecular assemblies inside living cells are pivotal in orchestrating various cellular functions and processes. However, designing such rhythmic patterns in synthetic cells remains a challenge. Here, we demonstrate the spontaneous emergence of spatio-temporal oscillations in the number of droplets, size, and their spatial distribution within a synthetic cell. The coacervate-based droplets in these synthetic cells sediment and fuse at the cell's bottom. Through a size control mechanism, the sedimented, large droplets shrink by expelling droplet material. The expelled molecules nucleate new droplets at the top of the synthetic cell, which grow and sediment again. These oscillations are sustained by converting chemical fuel into waste and can continue for hundreds of periods without evidence of fatigue. Strikingly, the period of the oscillation is in the minute's regime and tunable. The design of oscillating artificial organelles in synthetic cells brings us closer to creating more life-like materials and de novo life.

Oscillations in cells are essential for precisely regulating cellular and physiological processes in space and time[1]. Oscillations are not limited to fluctuation in concentrations of biomolecules but can also involve entire biomolecular assemblies[2]. An example is the Min oscillatory system in E. coli cells that regulates cell division using the Min-protein complex that oscillates from pole to pole[3]. Systems chemistry aims to mimic and engineer systems with the sophistication and complexity of biology in synthetic systems[4,5]. Engineered oscillations in synthetic cells could function as autonomous time-keeping systems that regulate the internal organizations of the synthetic cell. Nevertheless, designing oscillations in synthetic systems remains challenging. Several oscillators exist in which the concentration of components oscillate, so-called chemical oscillators. Classical, well-established oscillators like the Belousov-Zhabotinsky-reaction[6] or Liesegang patterns[7], and others[8] are based on inorganic compounds, making them incompatible with synthetic cells. Oscillators based on organic compounds can be designed using enzymatic reaction networks[9] and completely non-biological counterparts[10]. Self-assembling molecules can be coupled to these oscillating chemical reactions, like a pH-clock reaction[11], to produce oscillating assemblies like micelles[12], vesicles[13], or hydrogels[14]. Noteworthy, in such systems, the assemblies themselves follow the oscillatory behavior of the underlying reaction network and are thus themselves not responsible for the oscillations. This contrasts with most biological systems, like the min-protein complex, where the protein assemblies are directly responsible for the oscillations. Synthetic analogs of such systems, in which the assemblies are components directly responsible for the oscillating behavior, remain sparse[15–18].

Here, we developed synthetic cells endowed with active droplets as a model for synthetic membraneless organelles. When the cells are continuously supplied with chemical energy, they produce active droplets that spontaneously oscillate in their spatial distribution as well as number and size with a tunable period in the minute range. We found that the oscillations rely on a combination of a recently proposed size control mechanism[19,20], and fusion of droplets. Size control means these active droplets tend to evolve to an optimal size, making

[1]Department of Bioscience, School of Natural Sciences, Technical University of Munich, Lichtenbergstrasse 4, 85748 Garching, Germany. [2]Faculty of Mathematics, Natural Sciences, and Materials Engineering, and Institute of Physics, University of Augsburg, Universitätsstrasse 1, 86159 Augsburg, Germany. ✉e-mail: Christoph.Weber@physik.uni-augsburg.de; Job.Boekhoven@tum.de

droplets grow when too small and actively expel material when too large. When we inhibit droplet fusion, the average size and number of droplets are controlled, contrasting droplets in equilibrium, which would settle for one droplet per synthetic cell. In contrast, when fusion is possible, the fused, large droplets shrink at the bottom of the cell, which induces the nucleation of new droplets at the top of the synthetic cell, which grow and sediment again. The period of the oscillation is in the minute's regime and tunable. These oscillations are sustained by converting chemical fuel into waste and can continue for hundreds of periods without evidence of fatigue.

## Results

### Oscillations in synthetic cells

We prepared water-in-oil droplets as synthetic cells[21–23] that provide confinement while selectively exchanging material and energy with their surroundings (Fig. 1A). The synthetic cells are formed by combining an aqueous solution with a peptide and a polyanion with a perfluorinated oil that contains diisopropyl carbodiimide (DIC) as "fuel"[24]. Mixing the two solutions creates water droplets in the perfluorinated oil with a random size distribution between 10-50 μm in radius. The fuel crosses the interface to drive a chemical reaction cycle that produces active droplets[25–27], i.e., droplets made of transient droplet material[28–30]. Specifically, the fuel continuously activates AcF(RG)₃D-OH (peptide, Fig. 1B) and converts it into its corresponding cyclic anhydride (activated peptide, Fig. 1B). The activated peptide has a half-life of roughly a minute before it spontaneously deactivates through hydrolysis and reverts to its initial peptide state. The peptide is designed to be well-soluble in the synthetic cell. In its activated state, the peptide can form a complex coacervate with a polyanion (17 kDa polystyrene sulfonate, PSS). Consequently, the synthetic cells continuously produce transiently active material for active droplets at the expense of fuel that exchanges at the cell's interface, keeping the concentration of peptide, activated peptide, fuel, and waste in a steady state without further manipulation (Supplementary Fig. 1A-B and D-E). In previous work, we established that the peptide and fuel reside mostly in the aqueous phase outside these droplets, and thus, droplet material activation occurs predominantly outside[24]. The activated peptide partitions mostly inside the droplets, and thus, deactivation occurs predominantly inside the droplets. This asymmetry of activation and deactivation leads to an influx of activated droplet material and an efflux of deactivated droplet material over the interface. These fluxes are the prerequisite for the occurrence of oscillations.

We imaged the entire synthetic cell over time by obtaining Z-stacks in confocal microscopy (Fig. 2A, B, Supplementary Movie 1). We were surprised to find that the droplets showed oscillatory behavior—the droplets nucleated at the top of a synthetic cell, grew to a radius of roughly 1 μm, sedimented, and, at the bottom of the reactor, fused to form a few large droplets. Then, after about a minute, the process repeated, i.e., new droplets nucleated and sedimented. We measured the projected surface area of each droplet, from which we calculated the radius of each droplet with time. We quantified the total droplet amount in the synthetic cell's upper and lower half, which showed that oscillations in the lower and upper part are phase-shifted (Fig. 2C, Supplementary Fig. 2)—when the total droplet volume in the top half peaked, the bottom half's total droplet volume usually showed a valley and vice versa. Notably, the synthetic cells oscillated for hours without signs of fatigue (Fig. 2D).

The initial part of an oscillation can be explained intuitively. Complex coacervate droplets have a low interfacial surface tension, facilitating fusion[31]. Since their density is larger than their surrounding phase, droplets sediment[32,33]. To corroborate the effect of low interfacial tension and density-driven sedimentation, we encapsulated passive droplets—droplets that are not driven by a chemical reaction—and found that, in all cases, a single droplet was observed at the bottom of each synthetic cell (Supplementary Fig. 3). The next step is not intuitive, i.e., the large droplets at the bottom have to shrink to provide material for the renucleation of droplets at the top. Indeed, when we tracked the volume of a large droplet at the bottom of the reactor, we observed large droplets could shrink between fusion events (Fig. 2E). Such behavior is thermodynamically unfavorable since the decrease in droplet size and the renucleation of small droplets elsewhere increases the total droplet surface area and, thus, the free energy of the system. From this observation, we conclude that the droplet shrinkage in the last step of the oscillation cycle must be active behavior.

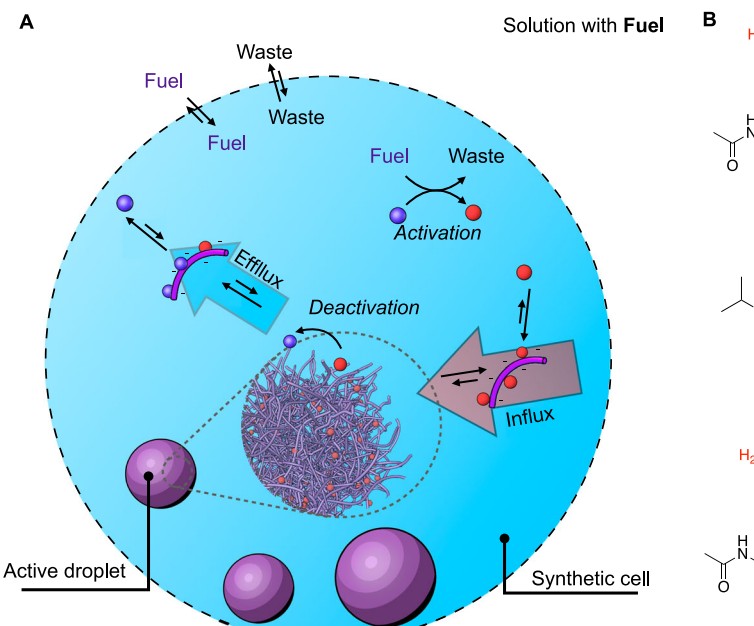

**Fig. 1 | Active droplets are maintained out of equilibrium in a synthetic cell. A** The experimental setup used to sustain droplets in a steady state. An aqueous droplet (synthetic cell) with peptide and polyanion is created in a liquid of perfluorinated oil with fuel. Fuel exchange at the oil-water interface leads to a steady state that produces chemically fueled droplets. **B** The chemical formulas of the reactants involved in the chemical reaction cycle.

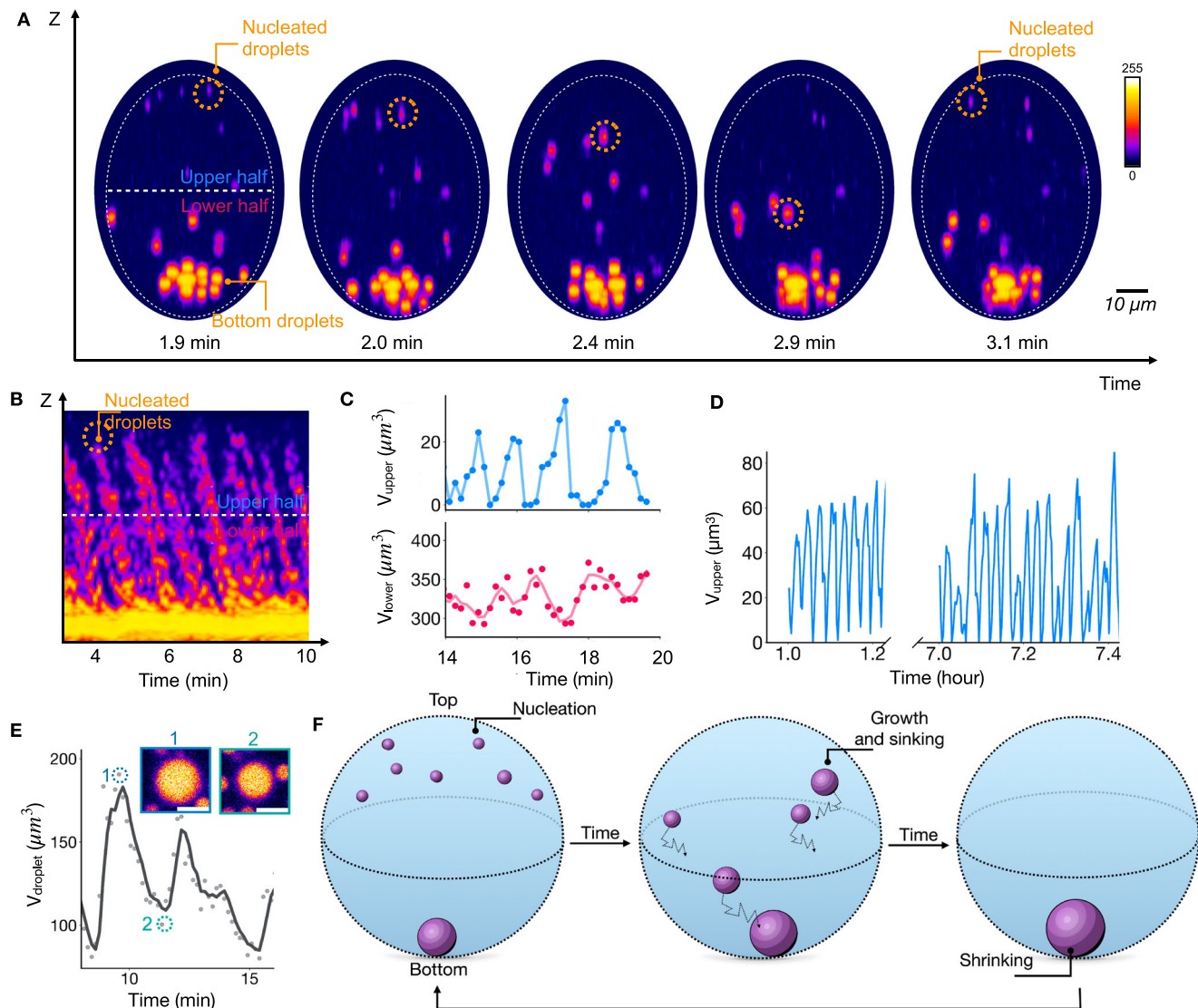

**Fig. 2 | Oscillations in the spatial distribution of artificial organelles in synthetic cells. A** Sideviews of a synthetic cell of radius 24.5 µm prepared from confocal Z-stacks. The synthetic cell contains 10 mM peptide, 5 mM PSS polyanion, and 200 mM MES buffer and is embedded in oil with 250 mM DIC as fuel. This fuel is continuously exchanged at the droplet-oil interface. The distribution of the artificial organelles oscillates with time. **B** Kymograph of the data from (**A**). **C** The volume of all artificial organelles combined in the upper or lower half of the synthetic cell as described in A and B. The markers represent the measured data, and the line represents the moving average of three data points. **D** The same as (**C**), but the data is displayed for several hours. Data corresponds to a different cell size ($r = 26.4$ µm), hence the difference in values (Supplementary Fig. 4) **E** The volume of a selected droplet at the bottom of the synthetic cell from (**A**). The two selected micrographs show the rapid shrinkage of the artificial organelle. Scalebars correspond to 5 µm. **F** Schematic representation of the three-phase oscillations: (1) Nucleation, (2) growth and sedimentation, (3) shrinkage. Source data are provided with this paper.

## Mono-disperse active emulsion: Active droplets evolve to the same optimal size

We tested whether the counterintuitive droplet shrinkage was a system property or a result of the confined environment of the synthetic cell. Thus, we prepared a mm-thick layer of the peptide solution and polyanion and carefully placed the pure diisopropyl carbodiimide as fuel on top (Fig. 3A). At the water-fuel interface, the fuel would exchange and saturate the aqueous buffer solution. We found the emergence of droplets throughout the thin water layer. The droplets fused rapidly and sedimented, increasing the average radii and decreasing the droplet number over time (Fig. 3B, Supplementary Fig. 5, and Supplementary Movie 2). Excitingly, tracking an individual droplet revealed that, between fusion events, the droplet's radius decreased, pointing to a combination of fusion and the droplet correcting its size by shrinking (Fig. 3B). This result demonstrates that the shrinkage of large droplets is independent of the confinement of the synthetic cell.

We reduced the complexity of the experiment by suppressing sedimentation and fusion—we performed the same experiments with 0.5% w/v agarose gel to suppress droplet fusion. After the fuel was carefully pipetted on top of the gel, droplets emerged, but no fusion was observed while the droplet number remained constant (Fig. 3C, Supplementary Fig. 6, and Supplementary Movie 3). Surprisingly, we tracked the radii of hundreds of droplets, which initially grew until they all stabilized at roughly 0.8 µm within 15 minutes (Fig. 3E and Supplementary Fig. 6A). We quantified the deviation of the mean radius to calculate the polydispersity index as a function of time (PDI = σ/µ, where σ is the standard deviation of the mean radius and µ is the mean radius), which settled at only 0.2 confirming a narrow radius distribution (Fig. 3F). Thus, the active droplets withstand fusion because of the gel network. Most importantly, they also withstand growth or shrinkage through Ostwald ripening, settling to a common optimal radius. We verified that the optimal radius of droplets was independent

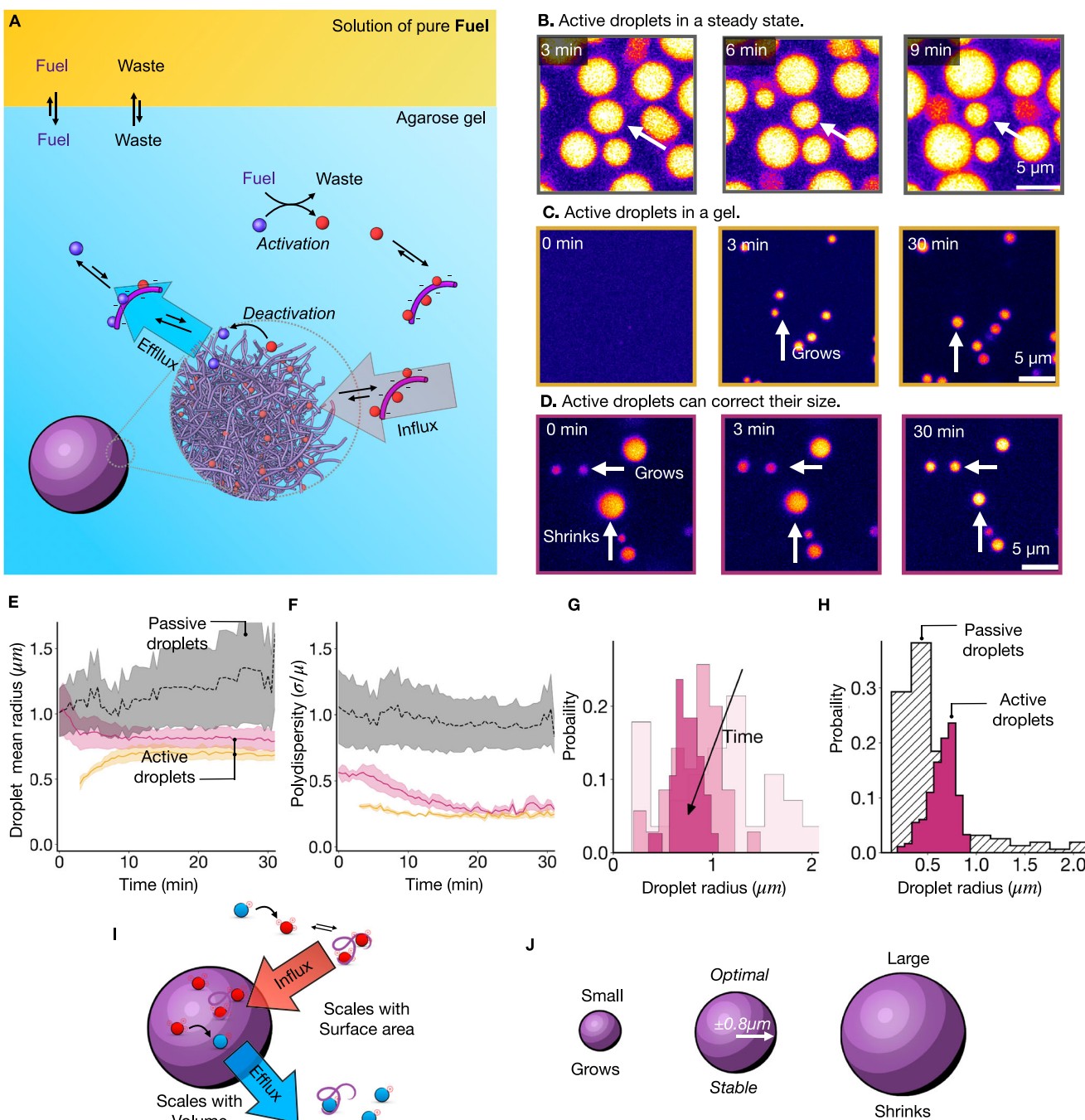

**Fig. 3 | Active droplets control their size. A** Schematic representation of the experiment. **B** Micrographs of active droplets continuously supplied with chemical fuel. A solution of 14 mM peptide, 15 mM PSS, 0.2 μM sulforhodamine B in 200 mM MES buffer at a pH of 5.3 was covered with a layer of DIC. The droplets were imaged by confocal microscopy. **C** Micrographs of active droplets under the same conditions in A, now embedded in a 0.5% agarose hydrogel to prevent fusion. **D** Micrographs of active droplets under the same conditions in C, now spiked with 20 mM EDC to induce pre-nucleation and fusion of droplets. **E, F** The average droplet radius (**E**) and polydispersity (**F**) for passive droplets (grey), active droplets trapped in a gel network (yellow), and pre-nucleated, active droplets trapped in a gel (purple). The active droplets evolve to the same stable radius, whereas the passive ones steadily grow. Lines represent the average for $N = 3$ and the shadowing the error as the standard deviation. **G, H** Histograms of the droplet radius distributions as a function of time for the experiment described in D (**G**) or for active and passive droplets in a gel after 60 min (**H**). **I** Schematic representation of the size control mechanism. Droplet material is activated outside of the droplet, leading to a surface area-dependent influx. Droplet material is deactivated inside of a droplet, which scales with its volume, leading to a volume-dependent efflux. **J** Surface area-dependent influx and volume-dependent efflux imply an optimal droplet radius. Small droplets will grow, and large droplets will shrink towards that radius. Source data are provided as a Source Data file.

of the agarose concentration to ensure the growth suppression was not a result of the gel's elasticity or mesh size (Supplementary Figs. 7–9)[34].

The droplets seem to grow to an optimal radius. To scrutinize this, we tested whether large droplets could also shrink to the same optimal size by initiating an experiment with a wide initial droplet size distribution, i.e., whether the emulsion could correct the size of droplets. We prepared active droplets using the water-soluble fuel 1-ethyl-3-(3-dimethylaminopropyl)carbodiimide (EDC) before the droplets were trapped in the gel network (0.5% w/v agarose gel), allowing for fusion.

Once the gel had been set, we again carefully pipetted diisopropyl carbodiimide on the gel as fuel to sustain the pre-nucleated droplets (Fig. 3D, Supplementary Movie 4). Tracking the droplet radii unambiguously showed the size control mechanism at work—we found droplets with a radius >0.9 μm shrunk while droplets with a smaller radius grew (Supplementary Fig. 10). Moreover, the average droplet size decreased to a stable size of 0.9 μm, and the initially wide size distribution with a polydispersity index of 0.45 dropped within 20 min (Fig. 3E, F). Notably, the experiments with or without pre-nucleated droplets converged to roughly the same radius and polydispersity (Fig. 3E, yellow and purple). Histograms of the droplet's radii distribution further corroborated the size control mechanism by gradually narrowing with time (Fig. 3G). As a final control, we prepared passive droplets, i.e., a mimic of our active droplets but not regulated by chemical reactions (See Materials and Methods). We prepared the droplets in a gel matrix, tracked their radii, and found they increased with time (Fig. 3E grey and Supplementary Fig. 11), likely through Ostwald ripening. We observed that the polydispersity index was higher than for active droplets (Fig. 3F grey, Fig. 3H). These observations confirm that only active droplets can evolve to a common, optimal radii.

## The mechanism of size control

We propose that the size control mechanism is related to a previously proposed mechanism using theory that was never demonstrated experimentally to the best of our knowledge[19]. When activation occurs predominantly outside the droplets, the droplet material's influx of activated droplet material scales with the droplet's surface area (Fig. 3I). When deactivation occurs predominantly inside the droplet, the efflux of deactivated droplet material scales with its volume. This interplay between surface area-dependent influx and volume-dependent efflux implies that large droplets (with a small surface area to volume ratio) will lose more deactivated droplet material than activated droplet material they take up—they shrink. In contrast, small droplets (with a large surface area to volume ratio) grow because they take up more material than they lose. Such a mechanism explains why all droplets converge to one optimal radius and why large droplets that have fused (e.g., in the oscillations) shrink.

To validate this mechanism, we used the previously determined chemical reaction cycle's kinetics (Supplementary Table 1, Supplementary Fig. 1) and the partitioning coefficient of the peptide, polyanion, and fuel in the droplets as well as the diffusion constant of the peptides in the droplet phase and estimated the diffusion constant of the fuel and the peptides in solution[24]. We developed a theoretical framework with all these parameters that simulate droplets' nucleation, influx, and efflux in a steady state like in our synthetic cells (Supplementary Method 2 and Supplementary Table 2). We initiated a simulation in which droplets could grow under concentrations of fuel and precursor, as in the experiments. The simulation showed that once a steady state was reached, the droplets reached an optimal droplet size, similar to the experimentally obtained values (Supplementary Fig. 12). Moreover, we redistributed the droplet material amongst those droplets in the final state of the simulation to create a perturbed emulsion—some droplets were larger, and others were smaller than the optimal radius (Fig. 4A). When we continued the simulation, the large droplets shrank while small ones grew, confirming the theoretically proposed size control mechanism.

This size control mechanism implies that synthetic cells can control the size and number of their synthetic organelles by investing chemical fuel in a chemical reaction cycle that activates droplet material in the dilute phase and deactivates it in the droplet phase. We prepared such synthetic cells using an artificial cytoskeleton of an agarose gel that contained precursor, polyanion, and buffer in a solution of perfluorinated oil with DIC. Droplets of similar radii were observed in the synthetic cell, and the total volume of all combined active droplets remained constant, pointing to a steady state (Supplementary Fig. 13). We varied the fuel concentration in the perfluorinated oil, which sets the steady-state concentration in the reactors (See Materials and Methods and Supplementary Fig. 1D and F). An increase in the steady-state concentration of fuel in the synthetic cell between 2.5 and 10 mM did not change the optimal droplet radii significantly (Fig. 4C). Instead, the droplet number density increased (Fig. 4B and D). Thus, more fuel produces more droplets of a similar size. In contrast, when we increased the peptide concentration in the solution, the droplet radii increased slightly while the droplet number density remained stable (Fig. 4E and F, respectively). Interestingly, we found that for each condition, the radii and their distribution width were independent of the synthetic cell size, even though increasing the size resulted in larger amounts of droplet material, which was accommodated by creating more droplets (Supplementary Fig. 4).

We used this theoretical model to probe these observations and predict the system's behavior outside of the range of parameters that we could experimentally vary (e.g., the deactivation rate constant or the concentration peptide beyond its solubility). The model confirmed the validity of the observation, i.e., how the concentrations of fuel and peptide set the optimal droplet radius and the number of droplets in the system (solid lines in Fig. 4C-F and Supplementary Fig. 14E-H). We also found that an increasing deactivation rate constant yields smaller droplets until no more droplets are found at a deactivation rate constant corresponding to a half-life of the activated peptide of 2.1 s (Fig. 4G). Biocatalysis can easily achieve this in the context of biomolecular condensates. It would require increased kinase activity in the droplets (deactivation) and increased phosphatase (activation) activity in the cytosol. Such a spatial separation is evidenced in P granules[35], stress granules[36,37], Cajal bodies[38]. Our work further adds to the notion that cells can control their organelle size by investing chemical energy in phase separation and spatially separating droplet material activation and deactivation. We find that the most critical parameters that influence the size of the droplets are the deactivation rate constant and the concentration of precursor.

Finally, we used the theoretical model to calculate a single droplet's net influx or efflux as a function of its radius (Fig. 4H). These calcualtions show that the net efflux increases drastically with increasing radius beyond the optimal radius of 0.75 μm.

## Size control and fusion drive oscillations

Both experiment and theory demonstrate that synthetic cells can control the size of their active droplets by balancing influx and efflux. This size control mechanism explains the counterintuitive part of the oscillations, i.e., sedimentation leads to droplets at the bottom of the synthetic cell, which enhances their fusion, leading to droplets with radii much greater than the optimal radius. Their large radii mean they expel large amounts of deactivated droplet material through the size control mechanism (Fig. 5A). We hypothesize that the expelled, deactivated material creates a concentration gradient from the cell's bottom to the top. At the bottom, near the large droplets, the deactivated peptide is reactivated and quickly reenters the large droplet. In contrast, at the top of the synthetic cell, the reactivated peptide cannot enter a droplet and thus accumulates. New droplets can only nucleate when sufficient reactivated peptide has accumulated at the top of the synthetic cell, i.e., when the concentration of activated peptide is above $c_{saturation}$. Then, after nucleation, the droplets continue growing while suppressing further nucleation events by locally lowering activated peptide concentration. Once droplets are sufficiently large to outcompete their Brownian motion, they sediment, leading to fusion at the bottom, restarting the oscillation.

Oscillations require non-linear steps to avoid dampening to a steady state[39]. In the proposed mechanism for the oscillation, several steps introduce these non-linearities. Specifically, the nucleation of new droplets at the top of the reactor requires to exceed a threshold

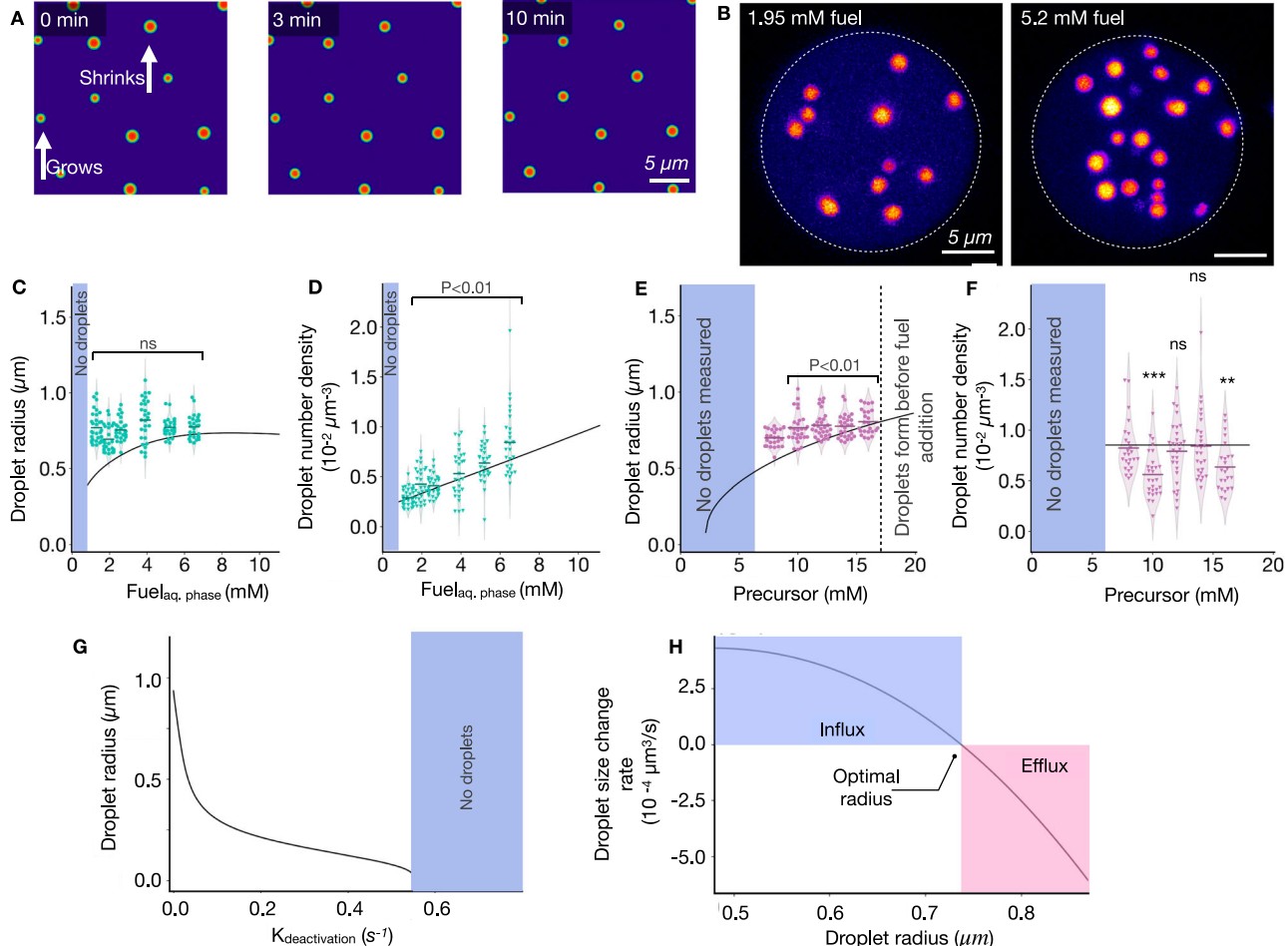

**Fig. 4 | Parameters that regulate the optimal droplet radii and droplet density.**
**A** Snapshots of the simulations of the model for active droplets (details in Supplementary Method 3) approaching their steady state after a size perturbation. When perturbing the steady state by increasing/decreasing randomly the size of droplets (left), decreased droplets grow, and increased droplets shrink toward their optimal radius (middle to right snapshot). **B** Z-projections of synthetic cells with an agarose-based cytoskeleton for two different fuel concentrations (1.95 mM left, and 5.2 mM right) indicate that more active droplets form when increasing fuel. The conditions are 14 mM peptide, 15 mM PSS, 0.2 µM sulforhodamine B in 200 mM MES buffer at a pH of 5.3. **C–F** The average droplet radius (**C–E**) and droplet number density (**D–F**) against the steady-state concentration of fuel (**C, D**) or peptide (**E, F**) agrees well with experimental data (dots) and theoretical model (solid lines). **G** The optimal droplet radius decreases with a faster deactivation rate $k_{deactivation}$. **H** The rate of change in the droplet radius vanishes at the optimal droplet radius. Above, droplets grow by influx of activated material, while below, they shrink by efflux of deactivated material. Source data are provided as a Source Data file.

concentration, the sedimentation of the droplets will only occur if they are sufficiently large and, finally, the size control mechanism scales non-linearly with the droplet's size (Fig. 4H).

Next, we investigated the parameters that affect the oscillations, starting with the fuel concentration. At 20 mM DIC in the oil phase, which corresponds to 0.26 mM according to $K_p$, no droplets were formed—according to the kinetic model, the activated peptide concentration is below the coacervation critical concentration of 0.025-0.05 mM (See Materials and Methods). In contrast, with 35 mM DIC in the oil phase, corresponding to 0.46 mM in the synthetic cell, led to the formation of a single stable droplet (Fig. 5E). When fuel was further increased to 70 mM (0.9 mM in the synthetic cell), we started observing oscillations. Specifically, we observed a single nucleation event at the top of the synthetic cell (Fig. 5B top, Supplementary Movie 5). This droplet sediments to the bottom and, finally, fuses with the big droplet, starting the cycle again. We observed that the number of droplets in the synthetic cell oscillated between one and two for the measured time. Indeed, the bottom droplet's volume peaked after fusion and decreased until the next fusion event. Because there was only one droplet nucleated, the measurements of droplet volume in the upper half of the container were much less noisy compared to the data in Fig. 2; compare Fig. 5C and D. Moreover, because there was only one

bottom droplet, we found that this droplet volume oscillates. We tracked the combined volume of all droplets in the synthetic cell and found that it also oscillates (Supplementary Fig. 15A-B).

When we increased the fuel concentration, the synthetic cells produced more droplet material (Fig. 5B bottom, Supplementary Fig. 17). Moreover, with each wave, more droplets nucleated simultaneously at the top of the synthetic cell, leading to larger droplets at the bottom after fusion. In such cases, the number of droplets becomes noisy (Supplementary Fig. 15C) due to the stochastic nature of fusion events. Nonetheless, the signal corresponding to the volume of droplets in the upper half clearly oscillated for all studied conditions (Fig. 5C, D). Finally, we noted the period decreased with increasing fuel (Fig. 5D, E).

Additionally, we also studied the effect of the size of the synthetic cell. Like varying the fuel concentration, we observed a threshold for oscillations—synthetic cells with a radius <16 µm display stable droplets without oscillations. The period decreased rapidly as we increased the radius to 40 µm (Fig. 5G). We found that the lag to nucleate new droplets also decreased, i.e., when the last droplets were fusing, the next wave had already nucleated (Fig. 5H). The time this process took seemed to set the lower limit of the oscillation period. To understand the relation between fuel concentration or cell size and the

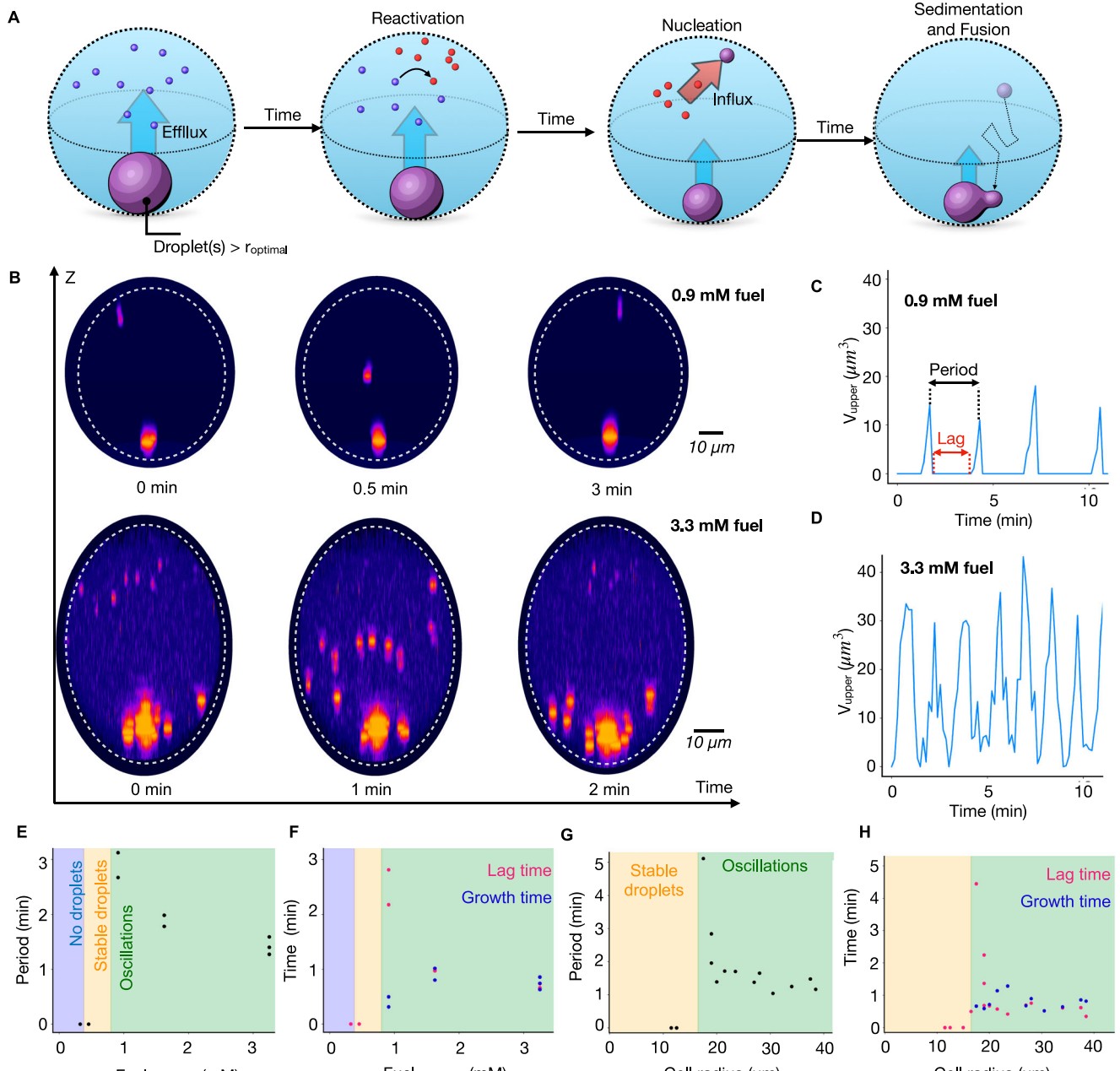

**Fig. 5 | Parameters that control the oscillation.** The conditions were 10 mM precursor, 5 mM PSS in 200 mM MES pH 5.3 with 0.5 μM sulforhodamine B as the dye, and 250 mM DIC was added to the oil phase to induce coacervation, except indicated otherwise. **A** Schematic representation of the events in each oscillation. New droplets nucleate and grow in the upper half of the reactors. Upon growth, these droplets sink down and fuse at the bottom of the reactor while new droplets nucleate in the upper half of the reactor. Grey dotted circles represent the periphery of the reactors. **B** Snapshots at different times of the side projections of two similarly sized cells when exposed to two different fuel concentrations, 0.9 mM (upper row, $r_{cell} = 25.6$ μm) and 3.3 mM (lower row, $r_{cell} = 24.5$ μm). The time

evolutions show that the number and the frequency of new droplets waves depends on fuel concentration. Synthetic cells appear elongated due to imaging artifacts. **C** The combined volume of all the artificial organelles in the upper half at 0.9 mM fuel and (**D**). at 3.3 mM fuel. **E** Period of the oscillations calculated from the dominant frequency determined by FFT as a function of fuel and (**G**). cell size. **F** Lag time between waves of nucleation (pink) and growth time (blue) at different fuel concentrations and **H** cell radius. In both cases, the lag time follows the same trend as the period, suggesting that the lag strongly influences the frequency of the oscillations. Source data are provided as a Source Data file.

period of the oscillations, we determined the lag period between the droplets sinking into the bottom part of the cell and the nucleation of new droplets at the top of the synthetic cell (See, for example, Fig. 5C). We found that it was as long as 3 min for low fuel levels and rapidly dropped with increasing fuel. For example, with 3.3 mM of fuel, the lag period was <40 s, meaning that new droplets nucleated before the previous wave completely fused. Similarly, the lag could range from several minutes for small synthetic with radii below 20 μm to very

short for synthetic cells with radii above 25 μm (Fig. 5H). Additionally, we calculated the time in which droplets were present in the upper half of the synthetic cell− the growth time−and found that it rapidly stabilized (Fig. 5F and H). Thus, close to the threshold for oscillations, the period of the oscillations is predominantly determined by the lag phase between droplet settling and re-nucleation of droplets, which is tuneable by the reactor size and the fuel concentration. In contrast, farther away from the threshold for oscillations, growth and lag time

approximately contribute equally to the oscillations period (Fig. 5F and H).

With our understanding of the mechanism of the oscillations from a chemical point of view, we focused on the influence of fusion and sedimentation rates. We increased the synthetic cell's viscosity by adding small amounts of agarose. We hypothesize that the increased viscosity decreases both fusion and sedimentation rates. Indeed, the droplets at the bottom remain smaller—reducing the droplet material efflux—and droplets remained at the upper half longer. A reduced efflux, decreased sedimentation, and fusion rates should lead to slower oscillations. Indeed, we found that the period of the oscillations increased (Supplementary Fig. 16).

## Discussion

We demonstrated a size control mechanism that sets the size of active droplets in an emulsion to an optimal size. This mechanism counteracts the thermodynamic driving forces that favor larger droplets to reduce interfacial surface tension. Our active emulsion continuously consumes fuel and uses its chemical potential to create many thermodynamically unfavorable droplets at a specific size. Our theoretical framework shows the behavior is generic, provided the emulsion fulfills a set of requirements. First, the droplet material must be coupled to a fuel-driven activation and deactivation reaction. In our emulsion, we used the carbodiimide-driven reaction cycle. Still, similar behavior could also be achieved by phosphorylation and dephosphorylation as described in synthetic emulsions[29,40] and biological ones using posttranslational modifications[41]. Second, the fuel-driven activation and spontaneous deactivation should occur simultaneously but spatially separated—activation occurs predominantly in the droplet, while deactivation occurs in the dilute phase. In our emulsion, this is achieved by a depletion of peptide in the droplets and an upconcentration of active peptide in the droplets. However, this could also be achieved by partitioning catalysts, such as a kinase in the dilute phase and phosphatase in the droplet phase. Third, the activation and deactivation rate constants are required to ensure enough droplet material is formed in a steady state and that the droplet material has a sufficiently long lifetime to induce droplet formation. Finally, the size control mechanism can inhibit Ostwald ripening, but under our conditions, the droplet size cannot be corrected if fusion occurs frequently. Thus, a network like a cytoskeleton is needed to prevent fusion. We believe this behavior could be generic based on these relatively simple requirements. We wonder if a cell uses similar active mechanisms to control the size of its membraneless organelles. Indeed, cells can control the size of their organelles[42], but the exact mechanisms remain elusive[43].

When fusion is not inhibited, and the volume of the emulsion is small, like the size of a cell, we show that the emulsion can spontaneously start oscillating between periods of fusion and periods of droplet shrinkage and renucleation of new droplets. These oscillations are driven by two mechanisms that regulate droplet size and work against each other, i.e., the size control mechanism shrinks large droplets while droplets grow through fusion as they sediment to the bottom of the synthetic cells. Our findings reveal that large sedimented droplets shrink by expelling droplet material, which nucleates new droplets at the top of the synthetic cell, sustaining the oscillations through repeated cycles. This oscillatory behavior can continue for hundreds of periods without fatigue and is tunable within the minute range.

The implications of our findings extend beyond the immediate study of oscillations in synthetic cells. The ability to design and control rhythmic patterns in synthetic cells opens alternative avenues for advancing synthetic biology and systems chemistry. Besides, our size control mechanism in a synthetic system offers a powerful model for biomolecular condensates in which such mechanisms are suspected to play a role.

## Methods

### Materials

Reagents and buffers were purchased from Sigma-Aldrich and used without further purification unless otherwise indicated. HPLC-grade acetonitrile (ACN) was purchased from VWR. The peptide Ac-F(RG)$_3$N-NH$_2$ (99%) was purchased from CASLO Aps. 2 w% 008-Fluorosurfactant in Novec 7500 was purchased from RAN Biotechnologies. Novec 7500 was purchased from Iolitec.

### Manual solid-phase peptide synthesis under controlled heating

AcF(RG)$_3$D-OH was synthesized on a 0.25 mmol scale using aspartic acid preloaded Wang resin (0.25 mmol, 0.67 mmol g$^{-1}$). The reaction vessel was connected to a nitrogen line, and the waste flask to a water pump. The resin was swell in DMF for 30 min with medium N$_2$ bubbling at room temperature. The peptide synthesis was performed at 68 °C.

Before each coupling step, the N-terminal Fmoc-protecting group was cleaved using 5 mL of a 0.2 M HOBt 5% (w/v) solution of piperazine in DMF. The reaction mixture was stirred with an N$_2$ stream for 1 min and 5 min. The deprotecting solution was removed, and the resin was washed with DMF. For each coupling, 3 eq. of amino acid were used together with 2.8 eq. of HCTU and 6 eq. of DIPEA. HCTU and DIPEA were added to the amino acid and vortexed until dissolved. The coupling solution was added to the resin and stirred with an N$_2$ stream for 7 min. After the coupling, the solution was drained, and the resin was washed with DMF. The deprotection, washing, coupling, and washing cycle was carried out for each amino acid. After the last coupling, the peptide was acetyl end-capped at room temperature. Therefore, a final deprotection was conducted; the resin was washed at room temperature, and 6 eq. of acetic anhydride and 6 eq. of DIPEA in DMF solution were added to the resin and stirred with N$_2$ stream for 10 min at RT. The resin was washed with DMF and DCM. A cleavage solution consisting of 2.5 % MQ-water, 2.5 % TIPS, and 95 % TFA was prepared to cleave the peptide from the resin. It was added to the resin and agitated for 2 h at RT. The cleavage solution was collected by filtration, and the resin was washed with DCM. The solvents were removed by co-distillation under reduced pressure using a rotary evaporator (Hei-VAP Core, VWR). The crude peptides were dissolved in H$_2$O:ACN (80:20) and purified using a reversed-phase preparative HPLC (Agilent 1260 Infinity ll setup, Agilent InfinityLab ZORBAX SB-C18 column 250 mm × 21.2 mm, 5 μm particle size, linear gradient of ACN from 2% to 98% and water, both with 0.1% TFA, flow rate 20 mL min$^{-1}$, retention time = 6 min)

The peptide was lyophilized (Christ Freeze Dryer Alpha 2–4 LDplus, VWR) and stored at −20 °C. It was characterized by electrospray ionization mass spectrometry (ESI-MS, Thermo Scientific, LCQ Fleet ION Trap Mass Spectrometer) in positive mode (Mass calc: 963.05 g/mol; Mass observed: 963.30 [Mw]$^+$), analytical HPLC (Thermofisher Ultimate 3000, 250 mm × 4.8 mm Hypersil Gold C18 column, linear gradient of acetonitrile 2–98% and water with 0.1% TFA) and $^1$H NMR. Characterization data was in agreement with already published data[20].

### Kinetic model

We used a kinetic model to predict the evolution of the product concentration over time and to model the steady-state product concentration at different conditions previously reported by Bergmann et al.[19] for the same peptide and carbodiimide. Supplementary Method 1 describes the model briefly. The rate constants used in this work are shown in Supplementary Table 1.

### Steady-state kinetics HPLC sample preparation

200 μL of the aqueous phase was prepared by adding the appropriate volumes of the precursor stock solution and diluting them in 0.2 M MES buffer pH = 5.3 to achieve a final concentration of 8–16 mM precursor. To simulate the gel layer system, 200 μL of pure DIC were added on top and the aqueous phase was gently stirred. To mimic the

synthetic cell container experiments, 1 mL of oil phase was prepared by adding DIC to perfluorinated oil (Novec 7500) to a final concentration of 50–500 mM. Then, the aqueous phase containing the precursor (200 µL) was added on top and the solution was vigorously stirred to form an emulsion. In both cases, 100 µL of the aqueous solution was pipetted from the solution and mixed with 100 µL of amylamine 200 mM to stop the reaction. Next, we performed an extraction step by adding 200 L of chloroform to remove DIC, which facilitates HPLC analysis. Finally, the aqueous phase was collected and injected without further dilution. All data points correspond to independent samples and were performed in triplicate ($N = 3$).

### Nuclear magnetic resonance spectroscopy (NMR)

$^1$H-NMR spectra were recorded with 16 scans at room temperature on a Bruker AVHD 300 spectrometer. Chemical shifts are reported in parts per million (ppm) relative to the signal of the reference compound TMPS ($\partial = -0.017$ ppm). All measurements were performed in duplicate ($N = 2$) at 25 °C. Concentrations were determined from the peak integrals, which were compared with the TMPS reference. The chemical shifts of the compared signals were $^1$H NMR (300 MHz, TMPS): TMPS $\delta$ (ppm) = 0 (s, 9H, CH$_3$), DIC $\delta$ (ppm) = 1.26-1.14 (d, 12H, CH$_3$), DIU $\delta$ (ppm) = 1-1.12 (d, 12H, CH$_3$).

### Steady-state kinetics NMR sample preparation

400 µL of the aqueous phase containing 14 mM precursor and 50 mM TMPS in 0.2 M MES pH 5.3 were prepared by dissolving the dry powder of TMPS and diluting the precursor stock. To start the reaction, 400 µL of DIC were added on top and the system was let to evolve for a certain amount of time. After that, 300 µL of the aqueous phase were mixed with 300 µL 640 mM borate buffer pH 10 containing 20% D$_2$O to quench the reaction by vortexing. All datapoints correspond to independent samples and were performed in duplicate ($N = 2$).

### Determination of DIC K$_p$

To determine the partition coefficient between the perfluorinated oil and 0.2 M MES buffer, pH 5.3 we dissolved 50 mM TMPS as reference in 500 µL buffer. 2.5 mL of perfluorinated oil containing 0.15-1 M DIC were added on top and samples were stirred vigorously to sustain the emulsion in the absence of surfactant. After 4 h, the two phases were allowed to separate and 300 µL of the aqueous solution were quenched with 300 µL 640 mM borate buffer pH 10 containing 20% D$_2$O. DIC and DIU in the solution were determined by $^1$H NMR spectroscopy as described above. DIC concentration in the aqueous phase was plotted against DIC concentration in the oil phase to obtain K$_p$. Values are collected in Supplementary Fig. 1.

### Determination solubility of DIC in 0.2 M MES buffer at pH = 5.3

DIC solubility was determined by $^1$H NMR spectroscopy. Samples were prepared by adding 500 µL of DIC on top of 500 µL 0.2 M MES buffer pH 5.3 and stirring gently for 6 h. After, 300 µL of the aqueous phase were quenched with 300 µL of 640 mM borate buffer pH 10 20% D$_2$O and measured as described above. Values are collected in Supplementary Fig. 1.

### High-performance liquid chromatography (HPLC)

High-pressure liquid chromatography was performed using analytical HPLC (Thermofisher Ultimate 3000) with a Hypersil Gold C18 column (250 mm × 4.8 mm). Separation was performed using a linear gradient of acetonitrile (2% – 85%) and water with 0.1 vol% TFA, and the chromatogram was analyzed using detector at 220 nm. The data was collected and analyzed with the Chromeleon 7 Chromatography Data System Software (Version 7.2 SR4). All measurements were performed in triplicate ($N = 3$) at 25 °C. Details about the solvent gradient used to determine peptide and active peptide concentration are shown in Supplementary Table 4.

### Coacervation critical concentration determination

CCC was determined by absorbance. Samples consisting of 10-x peptide, 5 mM PSS-17, and x mM of a non-hydrolysable mimic of our product (product*, AcF(RG)$_3$N-NH$_2$) in 200 mM MES at pH 5.3 with x ranging from 0 to 0.2 mM. Samples were equilibrated for 30 min at 21 °C and its absorbance was measured at 500 nm. Absorbance values above 0.2 AU were considered turbid.

### Method to sustain steady-state in confinement

We produced surfactant-stabilized water in oil droplets using 1 w% 008-FluoroSurfactant in 3 M HFE7500 as the oil phase. To form containers of varying size, 5 µL of the aqueous solution were added to 50 µL of the oil phase in a 200 µL Eppendorf tube. Snipping of the centrifugal tube resulted in the formation of containers with a random size distribution. The microreactors were imaged at the confocal microscope in untreated observation chambers consisting of a 24 mm × 60 mm glass cover slide and a 16 mm × 16 mm glass cover slide that were separated by two slices of double-sided sticky tape and sealed with two-component glue. For experiments that require hindered fusion, the aqueous solution contained 8–16 mM precursor, 8.6–17 mM PSS and 0.2 µM sulforhodamine B in 0.5 w/v % agarose 200 mM MES at pH 5.3 and the oil phase contained 0–500 mM DIC. Both solutions were kept at 45 °C until they were mixed to ensure water in oil droplet formation before agarose gelation.

In free fusion experiments, the aqueous solution contained 10 mM precursor, 5 mM PSS and 0.5 µM in 200 mM MES at pH 5.3 and the oil phase contained 0–250 mM DIC.

In both cases, samples were imaged after 30 min to guarantee that the steady-state concentrations were already reached.

### Method to sustain steady-state in bulk

We prepared a layer of lower melting point (LMP) agarose gel (0.5 w/v % except otherwise stated) containing 14 mM precursor, 15 mM PSS and 0.2 µM sulforhodamine B by heating a 0.75 w/v % LMP agarose 0.2 M MES buffer pH 5.3 stock solution to 45 °C and adding the appropiate volumes of precursor, PSS and sulforhodamine B stock solutions, all in 0.2 M MES buffer pH 5.3. 15 µL of the resulting solution were transferred to an Ibidi µ-Slide 15 Well 3D Glass Bottom previously passivated with PVA while still warm and let to jellify for 10 min. This resulted in a gel layer of ~1 mm thickness, which ensured no fuel gradients. To start the reaction, we added 15 µL of DIC on top and started imaging immediately.

For prenucleated droplets, the same procedure was followed but 20 mM EDC were added to the aqueous solution while still liquid to trigger coacervate formation which allowed droplet fusion for a few minutes resulting in a wide size initial distribution. Next, the solution was transferred to a well and, after 3 min, the DIC layer was added and imaging started.

### Passive droplets preparation into a gel matrix

To study the effect of the gel on droplet size, we performed control experiments –passive droplets– where 1.4 mM of precursor were substituted by a non-hydrolysable mimic of our product (product*, AcF(RG)$_3$N-NH$_2$). In this case, a solution of 12.6 mM precursor, 1.4 mM product*, 15 mM PSS and 0.2 µM sulforhodamine B in 0.5 w/v % agarose 0.2 M MES pH 5.3 was prepared while the agarose solution was still warm and transferred to an Ibidi µ-Slide 15 Well 3D Glass Bottom previously passivated with PVA. Sample imaging was started after 3 min.

### Passive droplets preparation within synthetic cells

A solution of 9 mM precursor, 1 mM product*, 5 mM PSS and 0.2 µM sulforhodamine B in 0.2 M MES pH 5.3 was prepared as the aqueous phase. The oil phase did not contain any fuel. Synthetic cells were produced as described above.

## Confocal fluorescence microscopy

A Stellaris 5 confocal microscope with a 63x oil immersion objective (1.4 NA) was used to image the samples. We used sulforhodamine B as droplet dye, excited the fluorophore at 561 nm and imaged at 570–650 nm with a HyD detector. The pinhole was set to 1 Airy unit. Typically, pixel size was set to 103 nm and image size was set to 512 px x 512 px. For hindered fusion experiments in bulk, we imaged a 6 μm in thickness layer with a z-step of 0.36 μm and a time step of 30 s. The same z-step was applied to experiments under confined conditions, but z-size was modified to image the whole container. In free fusion experiments, time step was set to 9–11 s and z-step to 4 μm. All experiments were performed in triplicate ($N = 3$).

## Image analysis

All images were processed and analyzed using ImageJ (Fiji)[44]. First, a maximum intensity z-projection was obtained. Then, we applied a Gaussian Blur (sigma = 2 px) followed by thresholding with the Otsu algorithm. Next, we used the analyze particles plugin (size = 0.1 μm²-infinity, circularity = 0.8-1) to obtain the area and the xy position of each droplet at each timepoint as well as the droplet number in each frame. To finish, we used a modified version of Simple Tracker[45], based on MATLAB[46], to reconstruct each droplet trajectory using the Hungarian linker with 1 μm as maximum linking distance and a maximum gap of 3 frames. Droplet radius and volume was calculated from the area assuming a spherical droplet. Total volume of droplets was calculated as the sum of each droplet's volume. The kymograph was generated using ImageJ (Fiji). Therefore, a Z-stack time series was converted into an interpolated 3D projection along the Y-axis. The kymograph was obtained along a vertical line with a width of 400 pixels using the plugin KymographBuilder.

## Determination of the period, lag time, and growth time in oscillating synthetic cells

Wave parameters were determined using the total volume of droplets in the upper half of the cell over time. The signal was denoised using a rolling average (window = 3). The period was calculated from the dominant frequency which was determined by performing a fast Fourier transformation (FFT) using Scipy[47] in Python 3.4 after normalizing the signal to $\mu = 0$ and $\sigma = 1$. The lag time was measured as the time between the start and the end of a trough. The value was obtained from the average of all cycles for a given signal. Growth time was calculated by subtracting the lag time from the period.

## Statistics

Statistical analyses were performed using Excel. Experimental data were analyzed using a two-tailed $t$-test for two samples assuming unequal variances. $P < 0.05$ was considered statistically significant. The exact sample size n, alpha level α, and $P$-value for each test are given in the source data or Supplementary Information.

## Data availability

Source data are provided with this paper. Further data supporting this study's findings are available from the corresponding authors upon request.

## Code availability

The code used to generate the data is available at https://doi.org/10.24433/CO.7419838.v1.

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

## Acknowledgements

The BoekhovenLab is grateful for support from the TUM Innovation Network - RISE funded through the Excellence Strategy. This research was conducted within the Max Planck School Matter to Life, supported by the German Federal Ministry of Education and Research (BMBF) in collaboration with the Max Planck Society. Funded by the Deutsche Forschungsgemeinschaft (DFG, German Research Foundation) – Project-ID 364653263 – TRR 235, under Germany's Excellence Strategy - EXC–2094 – 390783311 and the European Research Council (ERC starting grant 852187). The Weber group acknowledges funding by the European Research Council (ERC) under the European Union's Horizon 2020 research and innovation program (Fuelled Life with Grant agreement No 949021).

## Author contributions

J.B. and C.W. designed and supervised the project. J.S and A.M.B. designed the experiments. J.S., A.M.B. and M.S. performed the experiments. J.S., A.M.B. and M.S. analyzed the data. A.T. performed the theoretical calculations. J.B., C.W., J.S, A.T. and M. T-S. wrote and edited the manuscript.

## Funding

## Competing interests

The authors declare no competing interests.
