## [Transparent Peer Review file · Nature Communications]

Size control and oscillations of active droplets in synthetic cells

Corresponding Author: Professor Job Boekhoven

Version 0:

Reviewer comments:

Reviewer #1

(Remarks to the Author)

I appreciate the authors' efforts to address my comments, run additional experiments, and modify the manuscript. The significance of the work is sufficiently discussed, and the conclusions are supported by the results. I am convinced the manuscript is suitable for publication in Nature Communications.

Reviewer #2

(Remarks to the Author)

Authors properly addressed by criticism. I support publication of the manuscript in the present form.

Reviewer #3

(Remarks to the Author)

The authors have performed some additional experiments and addressed many of my previous comments. The manuscript has been corrected and improved significantly, but the following comments must be addressed before I can recommend publication of this manuscript.

1. Supplementary figure 3b also shows oscillation, which should not be expected. Authors should clarify this. In Supplementary figure 3, why only single coacervate droplet was observed is not understandable. The data for V_{upper} , V_{lower} , V_{total} , is not provided for passive droplets.
2. There is a clear lack of positive control, which can be solved by imide in place of anhydride. I request the authors to perform this critically important experiment.
3. As the authors have mentioned in their response "... you are correct to say that the droplet's position is not oscillating back and forth, and we falsely created that impression by stating "oscillations in position". Therefore, to improve clarity, we have replaced the phrase "oscillation in position" with "oscillation in the spatial distribution" throughout the manuscript. I would recommend them to make the title clearer e.g. "oscillation in formation of droplets or oscillatory distribution of droplet".
4. The authors have acknowledged in their response "While differences in radius may be subtle—especially when comparing values like $300 \mu\text{m}^3$ and $350 \mu\text{m}^3$, which translate to only a slight change in radius", I would expect data sets to be presented in terms of radius and should be included in SI. They should acknowledge the same in the manuscript as well.
5. Did the authors not observe the formation of activated peptide in case of experiment corresponding to supplementary figure 21?
6. Replace the term organelle with synthetic or artificial organelle in SI and the manuscript.
7. "activation occurs predominantly in the droplet, while deactivation occurs in the dilute phase", this statement is not very clear.
8. Some panels in supplementary figure 15 do not have a unit for y-axis.
9. The condition for determining the solubility of DIC in water (vigorous vortexing) is completely different than that from the water-in-oil experimental condition (no vortexing or shaking). There should be a justification in the revised manuscript.
10. The reaction constants are not defined in supplementary table 1. It would be better to add a scheme there for an easy correlation.

Version 1:

Reviewer comments:

Reviewer #3

(Remarks to the Author)

The authors have taken care of almost all the comments. However, for the supplementary figures 2 and 3, where data in terms of radius is provided, still the y-axis is unnecessarily of wide range for fig 3. It is surprising to see the difference in y-axis ranges in these figures, especially when I found out that in their first revised version, based on my comment, the authors kept the range same for both active and passive system, yet in current version, they went back to the previous y-axis range! What I strongly suggest is, the range should be same in both the figures S2,S3 (active and passive) and axes should not be modified to make the comparison look better.

Minor comment, for the supplementary fig. 3c, the unit of radius is wrong.

REVIEWER COMMENTS

Reviewer #1 (Remarks to the Author):

I appreciate the authors' efforts to address my comments, run additional experiments, and modify the manuscript. The significance of the work is sufficiently discussed, and the conclusions are supported by the results. I am convinced the manuscript is suitable for publication in Nature Communications.

Thank you for the time you invested in the manuscript.

Reviewer #2 (Remarks to the Author):

The authors have performed some additional experiments and addressed many of my previous comments. The manuscript has been corrected and improved significantly, but the following comments must be addressed before I can recommend publication of this manuscript.

Thank you for the time you invested in the manuscript. We've addressed your comments below as far as we could.

1. Supplementary figure 3b also shows oscillation, which should not be expected. Authors should clarify this. In Supplementary figure 3, why only single coacervate droplet was observed is not understandable. The data for V_{upper} , V_{lower} , V_{total} , is not provided for passive droplets.

Below, we address each point in detail:

A. Oscillations in Supplementary Figure 3b:

We acknowledge that the data in Supplementary Figure 3b appears to show minor fluctuations in droplet size over time. These fluctuations, however, are within the experimental noise and do not represent true oscillations. Unlike the active droplets shown in Supplementary Figure 2, passive droplets lack a chemical reaction cycle, and their volume remains relatively stable, with a maximum variation of 7.5% over the observed time period (compared to the 28% observed in active droplets). We have clarified this in the figure legend to avoid any potential misinterpretation.

Revised legend caption for Supplementary Figure 3:

"B. Evolution of the droplet size (black) for a synthetic cell of radius 32.6 μm . For passive droplets, all droplet material fuses into a large droplet at the bottom. Accordingly, $V_{droplet} = V_{lower}$ (red) = V_{total} , and V_{upper} (blue) = 0. Minor fluctuations in volume (< 7.5%) are attributed to experimental noise. As no reaction cycle is present, droplet volume remains relatively stable over time, contrasting with the active droplets depicted in Supplementary Figure 2. C. The same data as in B, now showed as radius."

B. Observation of a Single Droplet:

As stated in the manuscript, coacervate droplets have low interfacial surface tension, which promotes rapid fusion. Additionally, their higher density relative to the surrounding aqueous phase causes them to sediment to the bottom of the synthetic cells. Consequently, regardless of the initial number of droplets, they fuse into a single droplet located at the bottom in the absence of a chemical reaction. To improve clarity, we have expanded the description in the main text:

Revised text excerpt:

"To corroborate the effect of low interfacial tension and density-driven sedimentation, we encapsulated passive droplets—droplets that are not driven by a chemical reaction—and found that, in all cases, a single droplet was observed at the bottom of each synthetic cell (Supplementary Figure 3)."

C. Data for V_{upper} , V_{lower} , and V_{total} :

We have included the data for V_{lower} (red) and V_{upper} (blue) in Supplementary Figure 3. For passive droplets, all droplet material fuses into a large droplet at the bottom. Accordingly, $V_{droplet} = V_{lower} = V_{total}$, and $V_{upper} = 0$. This information is consistent with Supplementary Figure 3 and is now explicitly stated in the revised figure legend:

"B. Evolution of the droplet size (black) for a synthetic cell of radius 32.6 μm . For passive droplets, all droplet material fuses into a large droplet at the bottom. Accordingly, $V_{droplet} = V_{lower}$ (red) = V_{total} ,

and Vupper (blue)= 0. Minor fluctuations in volume (< 7.5%) are attributed to experimental noise. As no reaction cycle is present, droplet volume remains relatively stable over time, contrasting with the active droplets depicted in Supplementary Figure 2. C. The same data as in B, now showed as radius.“

We hope this addresses the reviewer's concerns comprehensively.

2. There is a clear lack of positive control, which can be solved by imide in place of anhydride. I request the authors to perform this critically important experiment.

An imide would be a great positive control for our activated peptide. We have tried to synthesize it, but sadly, it is not possible at this moment. Instead, we use amidated asparagine (two amides instead of one imide). The amide mimics the activated state of the anhydride but cannot hydrolyze. It is thus a perfect positive control for the active peptide.

We used this to mimic the active droplet that cannot hydrolyze (we referred to that as passive control). They do not exhibit oscillatory behavior, proving that the reaction cycle is necessary for the oscillations.

3. As the authors have mentioned in their response "... you are correct to say that the droplet's position is not oscillating back and forth, and we falsely created that impression by stating "oscillations in position". Therefore, to improve clarity, we have replaced the phrase "oscillation in position" with "oscillation in the spatial distribution" throughout the manuscript. I would recommend them to make the title clearer e.g. "oscillation in formation of droplets or oscillatory distribution of droplet".

Thank you for the suggestions. We disagree that adding more detail to the title improves the clarity. Both reviewers 1 and 3 agree that we unequivocally demonstrated size control and oscillations of active droplets in synthetic cells.

4. The authors have acknowledged in their response "While differences in radius may be subtle—especially when comparing values like $300 \mu\text{m}^3$ and $350 \mu\text{m}^3$, which translate to only a slight change in radius", I would expect data sets to be presented in terms of radius and should be included in SI. They should acknowledge the same in the manuscript as well.

We have included the datasets in terms of radius in Supplementary Figure 2 and 3. For active droplets, Supplementary Figure 2C and D, the oscillations are still clear for both the r_{lower} and r_{total} . For passive droplets, Supplementary Figure 3C, it becomes even more clear the stable nature of the fused, sedimented droplet. However, we have chosen to present the data in terms of volume in the main text, primarily to distinguish between the sizes of individual droplets (expressed in radii) and the sum of multiple droplets (expressed in volume). We believe that volume provides a clearer representation in this context.

5. Did the authors not observe the formation of activated peptide in case of experiment corresponding to supplementary figure 21?

In the experiment corresponding to Supplementary Figure 21, we analyzed the sample by HPLC at time points 0 and 60 minutes to confirm the absence of ester formation. We observed all EDC is consumed, which is evidence for the peptide activation and deactivation. As noted in the manuscript, the activated peptide (anhydride) is highly unstable and does not persist once EDC is consumed. Consequently, there is no detectable formation of the activated peptide after this point.

6. Replace the term organelle with synthetic or artificial organelle in SI and the manuscript.

We have revised the manuscript and Supplementary Information to replace the term "organelle" with "synthetic organelle" throughout the text.

7. "activation occurs predominantly in the droplet, while deactivation occurs in the dilute phase", this statement is not very clear.

To improve clarity, we have revised the statement in the main text as follows:

"In our emulsion, this is achieved by a depletion of peptide in the droplets and an upconcentration of active peptide in the droplets. However, this could also be achieved by partitioning catalysts, such as a kinase in the dilute phase and phosphatase in the droplet phase."

8. Some panels in supplementary figure 15 do not have a unit for y-axis.

We have included the missing units in Supplementary Figure 15.

9. The condition for determining the solubility of DIC in water (vigorous vortexing) is completely different than that from the water-in-oil experimental condition (no vortexing or shaking). There should be a justification in the revised manuscript.

You are correct. To clarify this in the revised manuscript, we have added the following to the Materials and Methods section: "2.5 mL of perfluorinated oil containing 0.15–1 M DIC were added on top of the aqueous phase, and samples were stirred vigorously to sustain the emulsion in the absence of surfactant."

10. The reaction constants are not defined in supplementary table 1. It would be better to add a scheme there for an easy correlation.

We have added an additional row in the table with the reactions.

Reviewer #3 (Remarks to the Author):

Authors properly addressed by criticism. I support publication of the manuscript in the present form.

The authors have taken care of almost all the comments. However, for the supplementary figures 2 and 3, where data in terms of radius is provided, still the y-axis is unnecessarily of wide range for fig 3. It is surprising to see the difference in y-axis ranges in these figures, especially when I found out that in their first revised version, based on my comment, the authors kept the range same for both active and passive system, yet in current version, they went back to the previous y-axis range!

What I strongly suggest is, the range should be same in both the figures S2,S3 (active and passive) and axes should not be modified to make the comparison look better.

Minor comment, for the supplementary fig. 3c, the unit of radius is wrong.

Thank you for the time you invested in the manuscript. We have modified the y-axis in S3 to match the ones in S2 and corrected the units where wrong.